# Relationship between emotional labor and sense of career success among community nurses in China, Beijing: A cross-sectional study based on latent class analysis

Fengping Han[1], Aihua Li[2], Dongmei Zhang[3], Lanting Lv[4], Qian Li[5], Jing Sun[6]*

**1** Laboratory Center, School of Nursing, Peking University, Beijing, China, **2** Equipment division, AMHT Group, Aerospace 731 Hospital, Beijing, China, **3** Beijing Nursing Department, Dongcheng District Health Service Center, Beijing, China, **4** School of Public Administration, Renmin University of China, Beijing, China, **5** Editorial department, Chinese Journal of Modern Nursing, Beijing, China, **6** Community Nursing Department, School of Nursing, Peking University, Beijing, China

* sunjing99@bjmu.edu.cn

## Abstract

### Background

This study investigated different patterns of emotional labor among community nurses in China and analyzed the relationships between the sense of career success and emotional labor.

### Methods

A total of 385 community nurses from Beijing participated in this investigation. Latent class analysis was used to identify meaningful subgroups of participants, and analysis of variance was used to analyze relationships between emotional labor classes and the sense of career success.

### Results

Emotional labor among community nursing staff in China was divided into three latent classes: active (n = 90, 25.6%), apathetic (n = 65, 18.5%), and moderate (n = 197, 55.9%). The active emotional labor classes had significantly higher career success (p<0.05). The "gaining recognition" dimension showed significant differences across the three classes.

### Conclusion

Our findings suggested managers to implement a variety of measures to strengthen interventions for employees' emotional labor that are targeted to incentive mechanisms, which will improve nurses' sense of career success.

**Data Availability Statement:** All relevant data are within the paper and its Supporting Information files.

**Funding:** The author(s) received no specific funding for this work.

**Competing interests:** The authors have declared that no competing interests exist.

## Introduction

The concept of community nursing was first put forward in 1970, and described as the combination of public health and nursing work [1]. The district or community nurse is central to the healthcare of community residents [2]. Currently, community nursing work in China covers basic therapeutic care (70%) and public health services (30%) [3], such as inpatient and outpatient care, preventive healthcare, maternal and child healthcare, chronic disease management, health education, and family nursing [4]. The complexity of the community nursing working environment and the diversity of services creates higher requirements in terms of the knowledge and abilities of community nurses [5]. To meet the requirements of their job, community nurses need to provide high-quality mental and physical healthcare services, and also need to maintain the necessary emotional communication with service recipients to achieve performance satisfaction [6]. To understand patients' experiences and support relationships with patients, relatives, and colleagues, community nurses use emotional labor and control their emotional expressions [7]. Emotional labor is the third kind of labor after physical and mental labor, and refers to the process of psychological adjustment necessary to express the emotions expected by an organization [8]. Nurses should know and control the process required for managing their emotions and expressions [9]. Excessive emotional labor can cause mental health problems [10], musculoskeletal symptoms [11], cardiovascular disease [12], and other physical health problems [13]. Although emotional labor is an important part of nursing practice, it is usually invisible, neglected and under estimated [14]. And even literature shows that emotional labor exists across a broad spectrum of nursing work [15], only few researches on conditions and related factors of community nurses' emotional labor is available [16–18], and this topic remains to be investigated.

With the expansion of the scope of community nursing and improvement in quality requirements, the psychological pressure on community nurses has increased [19], along with issues associated with professional satisfaction [20] and career success [21]. Career success refers to the gradual accumulation or acquisition of work-related achievements and positive feelings by individuals throughout their work experience, including remuneration, promotion, recognition, and acceptance [22]. A previous study among nurses indicated that the process of applying emotional labor aimed to achieve organizational goals and improve nurses' salary, promotion opportunities, and recognition [23]. Previous studies have shown that emotional labor strategies have significant predictive effects on job stress, job satisfaction, and subjective job performance [24]. Researchers also found that nurses' career success was mainly affected by emotional labor [25]. How about the relationship between career success and emotional labor in community nurses is need to be explored.

Emotional labor is the multi-level and dynamic essence of emotional regulation [26], and includes surface acting (e.g., concealing negative emotion with a fake smile), deep acting (e.g., taking a positive and optimistic approach to eliminate negative emotions and conform to the requirements of the organization), and natural expression dimensions (e.g., the true expression of one's emotions in accordance with work requirements) [27]. The effect of emotional labor is the result of the mixed effect of these dimensions [28]. Previous research on emotional labor in nursing mainly centered on the degree of total and each dimension of emotional labor or their relationship with other occupational factors, and the results did not clarify the specific situation and characteristics of emotional labor for an individual [24,29,30]. Therefore, even if the scores for emotional labor measurement were consistent, internal heterogeneity could not be determined [31]. To solve this problem, it is necessary to use an individual-centered research method to analyze the heterogeneity of emotional labor among community nurses. Latent class analysis (LCA) focuses on the heterogeneity of individuals, and the latent class

classification of individuals can be determined according to the response pattern of an individual in the detection questions; this means the number and proportion in each category can be clarified [32]. Therefore, the level and internal characteristics of emotional labor of community nurses are a research issue that needs further discussion. Analysis of the potential categories of emotional labor among community nurses (from surface variable analysis to internal characteristic analysis) and the relationship between the level of emotional labor and the sense of career success were taken based on LCA. The results may be valuable for improving the emotional labor among community nurses, which may provide scientific data for improving the quality of community nursing services and promoting career success and team stability.

## Method

### Study design and sample

This study was a cross-sectional survey that conducted during the period from Sep 10 to Sep 30, 2020. The sample size was based on the basic proportion of the scale items, with one item corresponding to10 samples. The required sample size for this study was 350. Considering the possibility of a 10% sample loss (n = 35), the final sample size for this study was calculated at 385 community nurses. We recruited community nurses from 11 community health service centers in four districts in Beijing using convenience sampling. The inclusion criteria were nursing staff: (1) with a nurse's certificate; (2) pass the community nurse on-the-job training prescribed by health administrative department of province or city; (3) that had worked in the participating community health institutions for ≥1 year and (4) that provided informed consent for voluntary participation. Exclusion criteria were: (1) nurses that had returned to work after sick or casual leave <1 year previously; (2) nursing managers; (3) student nurses; and (4) nursing staff who were going to retire within 3 months.

## Materials and methods

### Subjects and procedure

Based on the principle of convenient cluster sampling, 11 community health service institutions in Beijing were selected. Nurses in selected communities who met the inclusion criteria were all participated in our study. The research described in this paper meets the ethical guidelines of the ethics committee of the Peking University Health Science Center. And it was approved by the ethics committee of the Biomedical Ethics Committee of Peking University (ref.no. IRB00001052-20052). Nursing department of each selected community institutions assisted to organize the nurses who participated in the survey to fill in the questionnaire together. Before the survey, members of the research team explained the purpose and process of the study to potential participants and individualized data were kept confidential. They were informed that the investigation process was anonymous, and they were free to withdraw at any time. If they wished to participate, an informed consent form was signed and they were included in this study. The members of the investigation group then sent out questionnaires to participants with instructions on completing the survey using unified guiding language. Participants completed the questionnaire by themselves, and members of the investigation team checked whether the questionnaire was complete. The questionnaires were completed in an office room environment.

### Measures

The research questionnaire comprised three parts: (a) general information questionnaire; (b) the Emotional Labor Scale (ELS); and (c) the Career Success Scale (CSS). The general

information questionnaire included 31 items such as gender, age, marital status, education level, work years, daily working hours, position, and continuing study. The ELS was developed by Diefendorff [33] and revised by Yin [34]. The three-dimensional sale comprises 14 items and is used to measure core characteristics of emotional labor: surface acting (seven items), deep acting (four items), and natural expression (three items). Responses are on a five-point Likert scale (1 = strongly disagree; 5 = strongly agree). A higher score indicates a higher level of emotional labor. In the present study, the Cronbach's alpha for the scale was 0.776.

The CSS [35] was developed from the career satisfaction scale. A Chinese version was developed by Li Ya [36]. The 21 questionnaire items are divided into four dimensions: career development (five questions), freedom and happiness (five questions), gain recognition (six questions), and international network (five questions). Responses are on a five-point Likert scale (1 = strongly disagree; 5 = strongly agree), with a total possible score of 105. The higher the score, the more likely the respondent is to be satisfied and successful. In the present study, the Cronbach's alpha for the scale was 0.951.

## Statistical analysis

Data were analyzed using SPSS version 20.0. Count data were described by frequency composition ratio, and data with a normal distribution were described by mean±standard deviation. Mplus version 7.0 was used to analyze the potential categories of emotional labor. Mplus software was developed by Linda Muthen and Bengt Muthen in 1998 and It is the most popular and powerful latent variable analysis software at present [37]. LCA model fit test indices included: Akaike information criterion (AIC), Bayesian information criterion (BIC), accommodated BIC, entropy value, Lo-Mendell Rubin (LMR) test, and bootstrap likelihood ratio test (BLRT) [38]. Smaller AIC and BIC values indicate a better simulation fit. Entropy ranges from 0 to 1, and represent the accuracy of model classification; entropy ≥0.8 indicates that the classification accuracy exceeds 90%. In addition, the fit differences of potential category models can also be determined by LMR and BLRT, and p-values <0.05 indicated the differences were statistically significant. The establishment of the optimal model considered the adaptation index and existing research and the interpretability of data results [39]. After the optimal model was chosen, the differences in sociodemographic variables in each class allocation were obtained using chi-squared tests. Analysis of variance (ANOVA) was used to analyze the influence of a sense of career success on classification of emotional labor. P-values <0.05 were considered statistically significant.

## Results

### Participants' characteristics

A total of 382 questionnaires were completed anonymously and 352 valid questionnaires were recovered (effective recovery rate: 96.8%). Table 1 shows participating community nurses' sociodemographic and occupational characteristics. Participants' mean age was 36.93±9.29 years and most were female (98.9%). Most participants (49.1%) had an associate degree, 45.2% has a bachelor's degree or above, and 5.7%graduate from secondary vocational schools. The average daily work duration was 8.11±0.68 hours, 20.5% (n = 72) of participants participated in continuing education, and 4.3% (n = 15) were engaged in scientific research. Community nursing work involved various roles such as nursing operations, family visits, chronic disease management, health education, prevention, and healthcare services.

**Table 1. Sociodemographic and occupational characteristics of community nurses (N = 352).**

| Variables | n | % |
|---|---|---|
| **Age** | | |
| <35 years old | 153 | 43.5 |
| ≥35 years old | 199 | 56.5 |
| **Gender** | | |
| Male | 4 | 1.1 |
| Female | 348 | 98.9 |
| **Marriage** | | |
| married | 279 | 79.3 |
| unmarried | 73 | 20.7 |
| **Highest educational level** | | |
| secondary vocational schools | 20 | 5.7 |
| junior college | 173 | 49.1 |
| undergraduate | 158 | 44.9 |
| postgraduate or above | 1 | 0.3 |
| **Income status** | | |
| 3000–5000 | 109 | 31.0 |
| 5000–7000 | 184 | 52.3 |
| 7000–9000 | 55 | 15.6 |
| >9000 | 4 | 1.1 |
| **Management work** | | |
| Yes | 71 | 20.2 |
| No | 281 | 79.8 |
| **Continuing education** | | |
| Yes | 72 | 20.5 |
| No | 280 | 79.5 |
| **Scientific research** | | |
| Yes | 15 | 4.3 |
| No | 337 | 95.7 |

## Description of classifications

The fit indices for the series of latent class models are presented in Table 2. All fit indices were considered, but the BIC value was emphasized according to the recommendation of a previous study [38]. The BIC value (12623.04) was lowest in the 3-class model and the LMR (0.0124) and BLRT (0.000) reached significant levels (P<0.05). In the 4-class model, the LMR (0.8961) was no longer significant and the BIC value showed no further increase. Therefore, the 3-class model was selected as the final model with high reliability.

**Table 2. Latent class analysis results for emotional labor among community nursing staff in Beijing.**

| Model | K | Log Likelihood | AIC | BIC | aBIC | Entropy | LMR | BLRT | Class probability |
|---|---|---|---|---|---|---|---|---|---|
| 1 | 56 | -6492.935 | 13097.87 | 13314.23 | 13136.58 | | | | |
| 2 | 113 | -6044.045 | 12314.09 | 12750.68 | 12392.20 | 0.915 | 0.0001 | 0.000 | 0.64/0.36 |
| 3 | 170 | -5813.111 | 11966.22 | 12623.04 | 12083.73 | 0.921 | 0.0124 | 0.000 | 0.26/0.18/0.56 |
| 4 | 227 | -5659.968 | 11773.94 | 12650.98 | 11930.85 | 0.945 | 0.8961 | 0.000 | 0.12/0.21/0.49/0.18 |

Note: p<0.05.

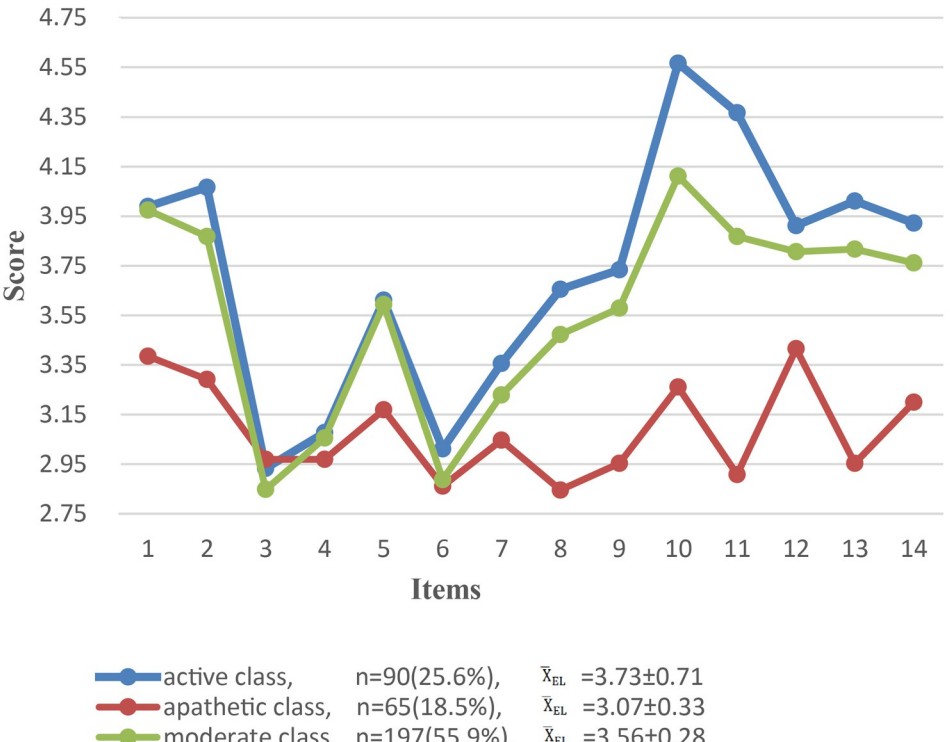

**Fig 1. Scores of the three latent classes for community nurses' Emotional Labor Scale items (N = 352).** Note: p<0.01.

Based on the LCA results, the 3-class model for community nurses' emotional labor is shown in Fig 1. Community nurses in class 1 had the highest scores for emotional labor, and 25.6% (n = 90) of participants belonged to this group. These nurses tended to score higher on most ELS items. Item 10 ("I work hard to feel the emotions that I need to show to customers") had the highest score. This indicated that these community nursing staff used a positive pattern to regulate emotional labor and keep their own emotions consistent with the requirements of the organization. Therefore, it was named the positive class.

Community nurses in class 2 exhibited different emotional labor patterns, and comprised 18.5% (n = 65) of participants. These nurses had the lowest ELS scores. The third item, "I put on a 'show' or 'performance' when interacting with customers," had the highest score of all classes. This indicated that these community nurses were good at using false emotions in their work. This lack of active use of emotional regulation meant that emotional labor was in a negative state. Therefore, this class was named the apathetic class.

The score curve for class 3 was basically the same as that for class 1. Community nurses in class 3 displayed medium ELS scores, and comprised 55.9% (n = 197) of participants. Community nurses belonging to class 3 were classified as the moderate class.

## Demographic and occupation characteristics by latent class

Table 3 presents participants' demographic and occupation characteristics by the three latent emotional labor classes. Professional title showed significant differences among the three class (p = 0.047). There were significant differences among the three emotional labor classes in different work places (p = 0.039) and for different work contents (Table 3). Good nurse-patient

**Table 3. Demographic and occupation characteristics by latent class (N = 352).**

| Variable | | Class1 No. | % | Class2 No. | % | Class3 No. | % | X² | p |
|---|---|---|---|---|---|---|---|---|---|
| **professional title** | | | | | | | | | |
| | Nurse | 25 | 27.8% | 23 | 35.4% | 37 | 18.8% | | |
| | Nurse Practitioner | 32 | 35.6% | 24 | 36.9% | 74 | 37.6% | 12.738 | 0.047 |
| | Supervisor nurse | 32 | 35.6% | 18 | 27.7% | 86 | 43.6% | | |
| | Professor of Nursing | 1 | 1.0% | 0 | 0.0% | 0 | 0.0% | | |
| **Working place** | | | | | | | | | |
| | Community service center | 54 | 60.0% | 49 | 75.4% | 114 | 57.9% | 6.481 | 0.039 |
| | Community service station | 36 | 40.0% | 16 | 24.6% | 83 | 42.1% | | |
| **Job content** | | | | | | | | | |
| Health education | | | | | | | | | |
| (No) | | 19 | 21.1% | 34 | 52.3% | 62 | 31.5% | 16.991 | ∠0.001 |
| (Yes) | | 71 | 78.9% | 31 | 47.7% | 135 | 68.5% | | |
| Chronic disease management | | | | | | | | | |
| (No) | | 42 | 46.7% | 44 | 67.7% | 115 | 58.4% | 7.108 | 0.029 |
| (Yes) | | 48 | 53.3% | 21 | 32.3% | 82 | 41.6% | | |
| Nursing procedures | | | | | | | | | |
| (No) | | 37 | 41.1% | 44 | 67.7% | 95 | 48.2% | 11.232 | 0.004 |
| (Yes) | | 53 | 58.9% | 21 | 32.3% | 102 | 51.8% | | |
| **Job satisfaction** | | | | | | | | | |
| | Good | 35 | 38.9% | 35 | 53.9% | 113 | 57.4% | | |
| | Common | 52 | 57.8% | 24 | 36.9% | 79 | 40.1% | 15.037 | 0.005 |
| | Poor | 3 | 3.3% | 6 | 9.2% | 5 | 2.5% | | |
| **Impact on service objects** | | | | | | | | | |
| | Yes | 67 | 74.5% | 32 | 49.2% | 129 | 65.5% | | |
| | Unclear | 20 | 22.2% | 26 | 40.0% | 57 | 28.9% | 11.439 | 0.022 |
| | No | 3 | 3.3% | 7 | 10.8% | 11 | 5.6% | | |
| **Nurse-patient relationship** | | | | | | | | | |
| | Good | 42 | 46.7% | 12 | 18.5% | 79 | 40.1% | | |
| | Common | 41 | 45.6% | 43 | 66.1% | 108 | 54.8% | 18.297 | 0.001 |
| | Poor | 7 | 7.7% | 10 | 15.4% | 10 | 5.1% | | |
| **Colleague relationship** | | | | | | | | | |
| | Good | 77 | 85.6% | 31 | 47.7% | 154 | 78.2% | | |
| | Common | 13 | 14.4% | 33 | 50.8% | 42 | 21.3% | 32.055 | ∠0.001 |
| | Poor | 0 | 0.0% | 1 | 1.5% | 1 | 0.5% | | |
| **Career interest** | | | | | | | | | |
| | Good | 55 | 61.1% | 19 | 29.2% | 98 | 49.7% | | |
| | Common | 27 | 30.0% | 43 | 66.2% | 91 | 46.2% | 21.486 | ∠0.001 |
| | Pool | 8 | 8.9% | 3 | 4.6% | 8 | 4.1% | | |
| **Career development** | | | | | | | | | |
| | Good | 56 | 62.2% | 25 | 38.5% | 114 | 57.8% | | |
| | Common | 28 | 31.1% | 37 | 56.9% | 76 | 38.6% | 12.179 | 0.016 |
| | Poor | 6 | 6.7% | 3 | 4.6% | 7 | 3.5% | | |

relationships and colleague relationships were significantly higher in the active class than in the other two classes (p<0.001). Community nurses in the active class had the highest job satisfaction (p = 0.005) and job relationships (p<0.001). Nurses in the active class had significantly

**Table 4. Correlation analysis between emotional labor and career success among community nurses (N = 352).**

|  | emotional labor | career success | career development | freedom and happy | gain recognition | international network |
|---|---|---|---|---|---|---|
| emotional labor | 1 |  |  |  |  |  |
| career success | 0.180** | 1 |  |  |  |  |
| career development | 0.100 | 0.846** | 1 |  |  |  |
| freedom and happy | 0.097 | 0.884** | 0.709** | 1 |  |  |
| gain recognition | 0.252** | 0.876** | 0.650** | 0.643** | 1 |  |
| international network | 0.171** | 0.908** | 0.672** | 0.745** | 0.762** | 1 |

*Note*: ** Correlation was significant at the 0.01 level (2-tailed).

higher levels of career interest (p<0.001) and positive attitudes about career prospects (p = 0.016) that the other classes.

The Pearson's correlation matrix between the total ELS score and total CSS score as well as for each dimension are shown in Table 4. In addition to the career development and freedom and happiness dimensions, the emotional labor of community nurses was positively correlated with occupational success (p<0.01).

The variance results are shown in Table 5. The latent subtypes were regarded as the independent variable and the CSS score was regarded as the dependent variable. The ANOVA results showed significant differences in the four career success dimensions between the active and apathetic classes (p<0.05). In addition, there was a significant difference (p<0.001) in the gain recognition dimension between the moderate and apathetic classes (C1>C3>C2).

## Discussion

This study used LCA to explore patterns of emotional labor among community nurses in China. The LCA method minimizes the differences within a category and maximizes the differences between categories and classifies group characteristics from the perspective of individuals; this approach clearly reveals difference between different heterogeneous individuals within the group [40,41]. Notably, we found there were distinct classifications of emotional labor in community nurses. The LCA results identified a 3-class model: class 1 = active group (n = 90, 25.6%), class 2 = apathetic group (n = 65, 18.5%), and class 3 = moderate group (n = 197, 55.9%).

The total mean emotional labor score was 3.51±0.49, which was higher than the medium critical value of 3.00 using the Likert scale 5-point scoring method [42]. This suggested that the emotional labor among community nurses was at a high level. It has also been reported that nurses' emotional labor is maintained at a medium to high level, such as in emergency treatment [43], the intensive care unit [44], and the neonatal ward [45]. Bolton [46] suggested that nursing is one of the occupations most commonly associated with extensive emotional work. Smith [47] highlighted that the nursing process (a framework for planning and implementing nursing care) involves person-centered rather than task-oriented care. With the increasing

**Table 5. Comparison of career success scale score by different emotional labor classes (N = 352).**

| variable | active class M±SD | apathetic class M±SD | moderate class M±SD | F | P | LSD |
|---|---|---|---|---|---|---|
| career development | 3.59±0.78 | 3.09±0.58 | 3.42±0.56 | 12.206 | <0.001 | C1>C2, C3>C2 |
| freedom and happiness | 3.61±1.05 | 3.13±0.72 | 3.49±0.72 | 7.077 | 0.001 | C1>C2, C3>C2 |
| gain recognition | 3.87±0.79 | 3.20±0.59 | 3.54±0.61 | 19.530 | <0.001 | C1>C3>C2 |
| international network | 3.71±1.00 | 3.13±0.65 | 3.49±0.67 | 10.838 | <0.001 | C1>C2, C3>C2 |

demand for high-quality healthcare service, medical organizations have emphasized provision of healthcare that centers on patients' needs. Furthermore, whether in the hospital or in the community, nurses need to comfort critically ill patients and their families, help them deal with illness correctly, and remove psychological barriers. All of which require the ability to exercise emotional labor. Thus, nurses' emotional labor is recognized as an integral part of patient care [17,48].

Community nurses in the active class displayed the highest total emotional labor score for all items except item 3 ("I put on a 'show' or 'performance' when interacting with customers"). In particular, item 10 ("I work at developing the feeling inside of me that I need to show to customers") showed the highest score. Most nurses agreed they overcame negative emotions at work, got along with others in a friendly way, and used deep acting emotional labor strategies, which means to keep inner feelings consistent with their displayed expression, instead of fake acting at work [49]. They may excel at actively regulating internal emotions and actively change to promote self-action, resulting in behavior consistent with organizational expectations [50]. Conversely, community nurses in the apathetic class displayed the lowest score for all items and the highest score of item 3. The highest score of item 3 in this class, may indicated that the apathetic class tended to use more fake emotions in work situations. They tried to hide their real emotional and facial expressions and show the emotion they expected to consisted with organization demand. Low emotional labor score may indicated that individuals in the apathetic class presented negative emotional manifestations and behaviors. Researchers found that emotional labor is usually stressful and has an adverse effect on nurses' psychological well-being and health, when emotions that are not genuinely felt have to be conveyed [51]. An overview of the effects of this disguise on the health revealed effects ranging from burnout and fatigue to dysmenorrhea, disruptions in sleep patterns and suicidal tendencies [52]. We also found the lowest scores were for item 8 ("I try to actually experience the emotions that I must show to customers") and Item 11 ("I work at developing the feeling inside of me that I need to show to customers"). Both of these items showed that nurses in the apathetic class did not want to "try to" or "develop" organizationally desired emotion expression during interpersonal transactions. A study found that if employees can actively internalize and integrate emotional performance norms and regard themselves as an important part of their work, their level of emotional labor motivation will be higher [53]. And previous researchers suggested to intervene by improving higher job autonomy, giving higher organizational identity and more support from superiors and colleagues to lower probability of mood disorders [8]. Class 3 was labeled the moderate group, the scores for emotional labor were in the middle range and the curves were basically the same as those for the active class. Previous studies proposed that financial rewards, genuine emotional expression and the maintenance of positive emotions, can make up for the mental resources consumed by emotional labor [54,55]. Grandey's study proved that employees in the supportive work environment of superiors and colleagues are more likely to have the tendency to conform to the expected emotion, and have the real emotional feelings of negative individuals to negative events [50]. Therefore, adjusting personal emotions and improving organizational management would be helpful to improve the enthusiasm of emotional labor of Class 2 and Class 3.

Our results showed significant differences in occupation characteristics between the three classes (Table 3). Community nurses' emotional labor was positively correlated with their job title, job contents, workplace, sense of job satisfaction, working relationships, career interest, and development [56]. Previous studies found that experienced workers have more mature emotional management skills [57]. Compared with a community health service center, health service stations have a smaller jurisdiction and population size, less contact between nurses and patients, lower workloads, and fewer interpersonal relationships to deal with. Therefore,

they are relatively prone to a negative emotional labor state [58]. Mróz et al. investigated working emotion among 55 employees and showed that positive emotional labor improved employees' job satisfaction [59]. A study reported that servers who expressed real smiles (didn't feel "false") at work, had more job satisfaction than those who reported faking emotions [60]. A study by Li Yongxin et al. [61] also showed positive relationship between active emotional labor and career interest. Previous studies also found that the degree of professional identity and job autonomy of nurses will affect their emotional labor ability [8,62,63]. Our study indicated that the active class of community nurses had the highest sense of career success, followed by the moderate class and then the apathetic group, and the gaining recognition dimension showed significant differences in the three classes. A study on preschool teachers also mentioned a link between emotional labor and a sense of career success [64]. Recognition refers to recognizing or paying special attention to employees' actions, efforts, behaviors, and performance, which is an important form of incentive for employees [65]. Herzberg's two-factor theory suggests that incentives can produce a sense of achievement, identity, and responsibility [66]. Yu's study also showed that professional identity, emotional labor, and job performance were positively correlated [67]. Above all, out study suggested to pay attention to employees' emotional labor skills training, negative emotions management and policy support and technical guidance provision to help boost positive emotions at work, which in turn increases job satisfaction and career success.

## Limitations

A limitation of this study was that it was a cross-sectional study rather than a longitudinal study. In addition, the sample size of this study was not large and widely enough, participants were nurses from 11 health service centers in four urban areas of Beijing, therefore, the representativeness and conclusions of its study were limited, and the results cannot represent all Chinese nurses. Further research may need to focus on nurses from national communities.

## Conclusion

The emotional labor of community nurses in China can be divided into three classes: positive, apathetic, and moderate. Managers could implement a variety of measures to strengthen interventions for employees' emotional labor that are targeted to incentive mechanisms, which will improve nurses' sense of career success.

## Supporting information

**S1 File.**
(DOCX)

## Acknowledgments

I would like to thank Ms. Jing Sun of the Peking University School of Nursing for her help in the research design and revision of the paper. I would like to thank my tutor Lanting lv an associate professor from Renmin University of China for her guidance on the design of research. Thank Dongmei Zhang, Aihua Li & Li Qian for their help in data collection and analysis.

## Author Contributions

**Investigation:** Aihua Li, Dongmei Zhang.

**Resources:** Fengping Han, Dongmei Zhang, Qian Li.

**Supervision:** Lanting Lv, Jing Sun.

**Writing – original draft:** Fengping Han, Qian Li, Jing Sun.

**Writing – review & editing:** Lanting Lv, Jing Sun.

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
