## [Decision Letter · Decision Letter 0]

25 Feb 2022

PONE-D-21-39368Relationship between emotional labor and sense of career success among community nurses in China, Beijing: a study based on latent class analysisPLOS ONE

Dear Dr. Sun,

Thank you for submitting your manuscript to PLOS ONE. After careful consideration, we feel that it has merit but does not fully meet PLOS ONE’s publication criteria as it currently stands. Therefore, we invite you to submit a revised version of the manuscript that addresses the points raised during the review process. The revised version should address all comments. Please submit your revised manuscript by Apr 11 2022 11:59PM. If you will need more time than this to complete your revisions, please reply to this message or contact the journal office at plosone@plos.org. Please include the following items when submitting your revised manuscript:A rebuttal letter that responds to each point raised by the academic editor and reviewer(s). You should upload this letter as a separate file labeled 'Response to Reviewers'.A marked-up copy of your manuscript that highlights changes made to the original version. You should upload this as a separate file labeled 'Revised Manuscript with Track Changes'.An unmarked version of your revised paper without tracked changes. You should upload this as a separate file labeled 'Manuscript'.

We look forward to receiving your revised manuscript.

Kind regards,

Petri Böckerman

Academic Editor

PLOS ONE

Journal Requirements:

Reviewers' comments:

Reviewer's Responses to Questions

**Comments to the Author**

1. Is the manuscript technically sound, and do the data support the conclusions?

Reviewer #1: Yes

Reviewer #2: Yes

2. Has the statistical analysis been performed appropriately and rigorously? 

Reviewer #1: I Don't Know

Reviewer #2: Yes

3. Have the authors made all data underlying the findings in their manuscript fully available?

Reviewer #1: Yes

Reviewer #2: Yes

4. Is the manuscript presented in an intelligible fashion and written in standard English?

Reviewer #1: Yes

Reviewer #2: Yes

5. Review Comments to the Author

Reviewer #1: Comments

General comment

The Manuscript is interesting and contributes a lot for better patient care

Abstract

Background: You should clearly state the research problem (Much better). The author starts with the research objective

Why this topic

What is the problem?

Method: Better to indicate your research design

Conclusion: Mention the major finding, then continue with your recommendation.

Result

Associate degree not clear try to clarify

Technical secondary school or below what does this mean? The participants were community nurses,

Explain how community nurses were recruited in your area somewhere in the methodology OR what is community nurses? What is their educational level in your setup?

Discussion and conclusion

Too long with unclear explanation

Try to choose major findings and discuss them by comparing them with the literature together with your assumption.

The manuscript should answer “So what”

What is the take-home message?

The manuscript has many terminologies which is difficult for readers, so better to have some operational definition.

Reviewer #2: Dear authors your study 'Relationship between emotional labor and sense of career success among community

nurses in China, Beijing: a study based on latent class analysis' was well done study.

Add the caption to the figure on page 27.

6. PLOS authors have the option to publish the peer review history of their article (what does this mean?). If published, this will include your full peer review and any attached files.

Reviewer #1: No

Reviewer #2: **Yes: **Mohammed Hasen Badeso

---

## [Author Response · Author response to Decision Letter 0]

7 Apr 2022

Response to Reviewers

Dear Reviewers,

On behalf of my co-authors, we thank you very much for giving us an opportunity to revise our manuscript. We appreciate reviewers very much for their positive and constructive comments and suggestions on our manuscript entitled “Relationship between emotional labor and sense of career success among community nurses in China, Beijing: a study based on latent class analysis” (PONE-D-21-54188). The revised version of our manuscript has been modified according to the reviewers’ suggestions.

The following is a point-to-point response to the editors’ and reviewers’ comments.

Reviewer #1: Comments

 Background: You should clearly state the research problem (Much better). The author starts with the research objective.

The research problem of our study is “The potential categories of community nurses’ emotional labor and related factors”. We made it clear in the Introduction part.

Why this topic 

We chose emotional labor as the study topic because excessive emotional labor can cause physical and mental health problems, job stress, burnout, et al. Although emotional labor is an important part of nursing practice, it is usually invisible, neglected and under estimated. So, this topic remains to be investigated. We made it clear in the Introduction part.

What is the problem

The problem is “what is the level and the internal characteristics of emotional labor of community nurses and its related factors”. We made it clear in the Introduction part.

Method: Better to indicate your research design

We added research design in the Method part of the manuscript, including inclusion criteria, sampling methods and distributing and retrieving questionnaires procedure, et al.

Conclusion: Mention the major finding, then continue with your recommendation.

We abbreviated the Conclusion part and leaving the major finding of the study.

Result

Associate degree not clear try to clarify. 

We revised the associate degree referred to the literature that used the same grouping method, and revised the Table 1. Hoping more clearly this time.

Technical secondary school or below what does this mean?

The meaning of “the technical secondary school or below” means nurses graduated from the secondary vocational school.

The participants were community nurses. Explain how community nurses were recruited in your area somewhere in the methodology OR what is community nurses? 

We added related information in the Method part to explain the criteria of the community nursing in China and how we recruited them.

Community nurse in China are nurses (with the nurse’s certificate) who pass the community nurse on-the-job training prescribed by health administrative department of province or city and working in the community health institution. 

Based on the principle of convenient cluster sampling, 11 community health service institutions in Beijing were selected. And community nurses in the 11 community health service institutions were potential participator of the study.

What is their educational level in your setup?

The education level of the community nurses in our setup was revised in the Result part: 5.7%graduate from secondary vocational schools, 49.1%participants had an associate degree, 45.2% has a bachelor’s degree or above. 

Discussion and conclusion

Too long with unclear explanation. Try to choose major findings and discuss them by comparing them with the literature together with your assumption.

We rewrote the Discussion part and made it clearer. And only the main findings were retained and we discussed our assumption by comparing with the related literature.

The manuscript should answer “So what”.

Yes, we rewrote the Discussion part of the manuscript and emphasized the scientific and practical significance of this study.

What is the take-home message?

The discussion section was rewritten to delete the long and irrelevant content, highlight the research focus, and reflect the key information. 

The manuscript has many terminologies which is difficult for readers, so better to have some operational definition.

We added operation definition or explanations of the terminologies in the revised version and hope it easier to be understood.

Reviewer #2: Dear authors mine study 'Relationship between emotional labor and sense of career success among community

nurses in China, Beijing: a study based on latent class analyses was well done study.

Add the caption to the figure on page 5418.

We added the caption of the Figure in the revised manuscript.

---

## [Decision Letter · Decision Letter 1]

25 Apr 2022

Relationship between emotional labor and sense of career success among community nurses in China, Beijing: a study based on latent class analysis

PONE-D-21-39368R1

Dear Dr. Sun,

We’re pleased to inform you that your manuscript has been judged scientifically suitable for publication and will be formally accepted for publication once it meets all outstanding technical requirements.

Kind regards,

Petri Böckerman

Academic Editor

PLOS ONE

Additional Editor Comments (optional):

Reviewers' comments:

Reviewer's Responses to Questions

**Comments to the Author**

1. If the authors have adequately addressed your comments raised in a previous round of review and you feel that this manuscript is now acceptable for publication, you may indicate that here to bypass the “Comments to the Author” section, enter your conflict of interest statement in the “Confidential to Editor” section, and submit your "Accept" recommendation.

Reviewer #1: All comments have been addressed

Reviewer #2: All comments have been addressed

2. Is the manuscript technically sound, and do the data support the conclusions?

Reviewer #1: Yes

Reviewer #2: Yes

3. Has the statistical analysis been performed appropriately and rigorously? 

Reviewer #1: Yes

Reviewer #2: Yes

4. Have the authors made all data underlying the findings in their manuscript fully available?

Reviewer #1: Yes

Reviewer #2: Yes

5. Is the manuscript presented in an intelligible fashion and written in standard English?

Reviewer #1: Yes

Reviewer #2: Yes

6. Review Comments to the Author

Reviewer #1: All comments have been addressed

The author carefully considered all comments and incorporated all appropriately.

Thank you for considering my comment

Reviewer #2: Very Good.

Well organized manuscript.

You Revised the manuscript as per the comments and all comments were addressed.

7. PLOS authors have the option to publish the peer review history of their article (what does this mean?). If published, this will include your full peer review and any attached files.

Reviewer #1: No

Reviewer #2: No

---

## [Editor Report · Acceptance letter]

28 Apr 2022

PONE-D-21-39368R1 

Relationship between emotional labor and sense of career success among community nurses in China, Beijing: a cross-sectional study based on latent class analysis 

Dear Dr. Sun:

I'm pleased to inform you that your manuscript has been deemed suitable for publication in PLOS ONE. Congratulations! Your manuscript is now with our production department. 

Kind regards, 

on behalf of

Professor Petri Böckerman 

Academic Editor

PLOS ONE